# MixCon3D: Synergizing Multi-View and Cross-Modal Contrastive Learning for Enhancing 3D Representation

## Abstract

By adapting the success of Contrastive Language-Image Pre-training from 2D vision to the 3D world, contrastive learning has emerged as a promising paradigm for open-world understanding jointly with text, image, and point cloud. While existing studies focus on aligning features from these individual modalities, this paper introduces a novel joint representation alignment approach. This mechanism enriches the conventional tri-modal representation by creating a new combined representation of images and point clouds, thus offering a more accurate depiction of real-world 3D objects and bolstering text alignment. The method, termed as MixCon3D, is furthered through the integration of multi-view images, offering a more holistic representation. Furthermore, we pioneer the first thorough investigation of various training recipes for the 3D contrastive learning paradigm, building a strong baseline with improved performance and generalizability. Extensive experiments conducted on three representative benchmarks reveal that our method renders significant improvement over the baseline, surpassing the previous state-of-the-art performance on the challenging 1,156-category Objaverse-LVIS dataset by **5.7%**. We further showcase the effectiveness of our approach in more applications, including text-to-3D retrieval and point cloud captioning.

## 1 Introduction

The ability to perceive and comprehend 3D environments is crucial in applications like augmented/virtual reality, autonomous driving, and embodied AI. Despite significant advancements achieved in 3D recognition (Qian et al., 2021; Wang et al., 2019b;a; Qi et al., 2017a;b; Qian et al., 2022; Yu et al., 2022; Lai et al., 2022; Zhao et al., 2021) , there is still a distinctive gap between the development of 2D and 3D vision methods. This phenomenon primarily stems from the limited diversity and complexity of existing 3D datasets caused by high data acquisition cost.

To tackle the data scarcity issue, recent research endeavors have turned to well-trained 2D foundation models. A line of such works is built upon CLIP (Radford et al., 2021), a pioneering foundation model known for its extraordinary zero-shot recognition capability on previously unseen objects by training on web-scale data. The knowledge learned from millions or even billions of image-text pairs proves to be invaluable in assisting the model to learn 3D shapes. In this context, ULIP (Xue et al., 2023a) and CLIP$^2$ (Zeng et al., 2023) first propose to keep the image and text encoder frozen, while training the 3D encoder on the (image, text, point cloud) triplets, which leads to substantially increased zero-shot 3D recognition performance.

While existing methods have demonstrated great promise, they predominantly center on a naive correspondence between point-text and point-image to form contrastive pairs, overlooking the intricate relationships across various modalities and perspectives. For instance, 2D images and 3D point clouds are known to capture distinct yet complementary attributes of an object (Bai et al., 2022; Liu et al., 2023b; Chen et al., 2023; Wang et al., 2023): point clouds emphasize depth and geometry, whereas images excel in representing dense semantic information. However, this synergy is underexplored, with each modality isolated in contrastive learning. Similarly, though multi-view learning (Su et al., 2015; Jaritz et al., 2019; Hamdi et al., 2021) has been extensively explored within the field of supervised 3D representation learning, it is also relatively underexplored within the con-

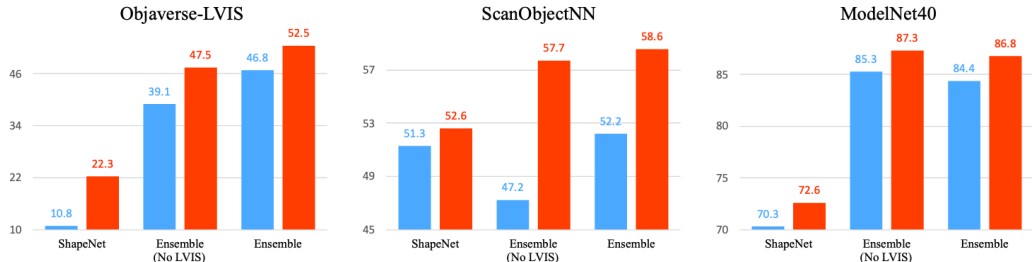

Figure 1: Comparison of zero-shot point cloud recognition between the OpenShape (red) and our MixCon3D (blue) under different pre-training datasets (ShapeNet, Ensemble (No LVIS) and Ensemble). Our model obtains consistent improvements on different types of training datasets on various downstream benchmarks.

trastive learning paradigm across multiple modalities. Moreover, while there has been exhaustive research on training protocols for 2D models, discussions on 3D training methodologies remain notably scarce.

To bridge these gaps, in this paper, we propose to synergize **Mult**i-view and **cross**-modal **CON**trastive learning, termed as **MixCon3D**, tailored to maximize the efficacy and potential of contrastive learning across images, texts, and point clouds. Central to our approach is a joint representation alignment technique, capitalizing on the complementarity between 2D images and 3D point clouds. By crafting a unified representation from these two modalities, MixCon3D offers a more holistic description of real-world 3D objects, enhancing its alignment with text through an added contrastive loss. Moreover, MixCon3D ensures a comprehensive capture of a 3D object by extracting features from multi-view images, thus fortifying cross-modal alignment. Through a careful examination of the training recipe (including batch size, temperature parameters, and learning rate schedules etc.), we establish an advanced training guideline. This not only stabilizes the training process but also drives enhanced performance.

As illustrated in Figure 1, our method consistently shows remarkable improvement over multiple popular 3D understanding benchmarks. On the well-established ScanObjectNN dataset, our approach surpasses the previous best method by 6.4%, indicating the generalization ability of MixCon3D. Moreover, on the challenging 1,156-category Objaverse-LVIS dataset with long-tailed distribution, our MixCon3D attains an accuracy of 52.5%, outperforming the competing models by a significant margin. Lastly, following OpenShape (Liu et al., 2023a), we demonstrate the newly learned 3D embedding space is well aligned with CLIP image and text embedding space by employing the learned 3D features of our model in text to 3D shape retrieval and point cloud caption generation tasks.

## 2    RELATED WORKS

**3D Representation Learning.**  Point-based methods, a prominent category of 3D representation learning methods, have garnered much attention for their simplicity, effectiveness, and efficiency. The pioneering work, PointNet (Qi et al., 2017a), models the inherent permutation invariance of points with point-wise feature extraction and max-pooling, enabling direct processing of unstructured point sets. PointNet++ (Qi et al., 2017b) enhances PointNet with a hierarchical network architecture to effectively capture both local and global geometric cues. Building upon this foundation, the 3D community has witnessed the emergence of a plethora of point-based methods, with a particular focus on the design of effective local modules (Qian et al., 2021; Wang et al., 2019b;a; Thomas et al., 2019; Tatarchenko et al., 2018; Xu et al., 2018; Liu et al., 2019; Zhao et al., 2021). PointNext (Qian et al., 2022) explores an orthogonal direction, underscoring the pivotal role of training and scaling strategies in effective 3D representation learning.

Another line of work focuses on designing self-supervised learning techniques tailored for point cloud understanding. Early endeavors along this direction centered around the proposition of various low-level pretext tasks, including self-reconstruction (Achlioptas et al., 2018; Deng et al., 2018),

distortion reconstruction (Sauder & Sievers, 2019; Mersch et al., 2022), and normal estimation (Rao et al., 2020). Recently, the remarkable success of self-supervised learning in the language and vision domain has prompted researchers in the 3D domain to adopt analogous self-supervised learning paradigms. PointContrast (Xie et al., 2020), for instance, leverages the concept of contrasting two views of the same point cloud to facilitate high-level scene understanding. PointBERT (Yu et al., 2022) and PointMAE (Pang et al., 2022), based on the idea of masked modeling, train an autoencoder to recover the masked portion of data with the unmasked part of the input.

Different from designing better 3D backbones or self-supervised learning pretext tasks, this paper focuses on improving multimodal contrastive learning for 3D open-world understanding.

**CLIP for 3D open-world understanding.** By training on vast web-scale image-text pairs, CLIP (Radford et al., 2021) has revolutionized the area of visual representation learning via language supervision. The extraordinary zero-shot recognition performance of CLIP has found applications in a lot of domains, including zero-shot text-to-3D generation (Hong et al., 2022; Jain et al., 2022; Michel et al., 2022; Sanghi et al., 2022), zero-shot 3D segmentation or detection (Jatavallabhula et al., 2023; Ding et al., 2023; Yang et al., 2023; Lu et al., 2023), and 3D shape understanding (Zhang et al., 2022; Zhu et al., 2022; Xue et al., 2023a;b; Qi et al., 2023; Hegde et al., 2023). The initial exploration in leveraging CLIP for 3D shape understanding typically involves the projection of the original point cloud into depth maps, followed by the direct application of 2D CLIP on them (Zhang et al., 2022; Zhu et al., 2022). However, this approach suffers from information loss during projection, while introducing extra latency. Plus, the domain gap between the synthetically rendered depth maps and natural images could significantly hurt CLIP performance. Some more recent works propose to learn a unified embedding space for image, text, and point cloud by training a 3D encoder aligned with CLIP image/text encoder (Xue et al., 2023a;b; Liu et al., 2023a). Our work follows this framework, but takes one step ahead to fully unleash the power of contrastive learning between (image, text, point cloud) triplets by 1) proposing an image + point cloud joint representation alignment mechanism; 2) utilizing multi-view images for better point-image alignment.

## 3 MIXCON3D

In this section, we detail our proposed MixCon3D. In Section 3.1, we first review the existing paradigm of image-text-3D contrastive training, *i.e.*, contrastive learning between three modalities of image, text and 3D point cloud. Then, we analyze and improve the existing training recipe, leading to a substantially stronger baseline in Section 3.2. Built upon the strong baseline, in Section 3.3, we propose a novel joint representation alignment mechanism, aggregating complementary information captured by 2D and 3D sensors before aligning with text features. Furthermore, in Section 3.4, we propose to leverage multi-view images to reflect the actual 3D world comprehensively and enhance the image representation quality.

### 3.1 PRELIMINARY: IMAGE-TEXT-3D CONTRASTIVE LEARNING

By exploiting a huge amount of image-text pairs crawled from the web, the CLIP model (Radford et al., 2021) has demonstrated exceptional open-world image understanding capability. Typically, given batched image-text pairs $\{(\boldsymbol{x}_i^I, \boldsymbol{x}_i^T\}_{i=1}^N$ and the (image, text) encoders $(f^I, f^T)$ and corresponding projection heads $(g^I, g^T)$, CLIP is trained by the contrastive loss $\mathcal{L}^{I \leftrightarrow T}$ as follows:

$$\mathcal{L}^{I \leftrightarrow T}(\boldsymbol{x}_i^I, \boldsymbol{x}_i^T) = -\frac{1}{2N} \sum_i^N (\log \frac{\exp(\boldsymbol{z}_i^I \cdot \boldsymbol{z}_i^T / \tau)}{\sum_j \exp(\boldsymbol{z}_i^I \cdot \boldsymbol{z}_j^T / \tau)} + \log \frac{\exp(\boldsymbol{z}_i^T \cdot \boldsymbol{z}_i^I / \tau)}{\sum_j \exp(\boldsymbol{z}_i^T \cdot \boldsymbol{z}_j^I / \tau)}) \quad (1)$$

where $\tau$ is a learnable temperature, and $(\boldsymbol{z}_i^I = g^I \circ f^I(\boldsymbol{x}_i^I) / ||g^I \circ f^I(\boldsymbol{x}_i^I)||, \boldsymbol{z}_i^T = g^T \circ f^T(\boldsymbol{x}_i^T) / ||g^T \circ f^T(\boldsymbol{x}_i^T)||)$ are the L2 normalized (image, text) features output by projection heads.

As the scale of 3D datasets is quite limited, previous works (Xue et al., 2023a;b; Liu et al., 2023a; Zeng et al., 2023) have resorted to the pre-trained CLIP image and text embedding space for training a native 3D model $g^P \circ f^P$ (including 3D encoder $f^P$ and projection head $g^P$) with open-world recognition ability. Since CLIP is pre-trained on a much larger data scale and much more well aligned, its image model $g^I \circ f^I$ and text model $g^T \circ f^T$ are frozen during training. Specifically, given batched $N$ input image $\boldsymbol{x}_i^I$, text $\boldsymbol{x}_i^T$, and point cloud $\boldsymbol{x}_i^P$ triplets $\{(\boldsymbol{x}_i^I, \boldsymbol{x}_i^T, \boldsymbol{x}_i^P)\}_{i=1}^N$ (hence

| Method | Temperature Parameter | Batchsize | Learning Rate Schedule | Warm up | EMA |
|---|---|---|---|---|---|
| ULIP | Share | 64 | Cosine Decay | ✓ | ✗ |
| OpenShape | Share | 200 | Step Decay | ✗ | ✗ |
| Improved Recipe | Separate | ∼2k | Cosine Decay | ✓ | ✓ |

Table 1: The summary and comparisons between the baseline and our improved training recipe.

the name image-text-3D), the 3D model $g^P \circ f^P$ is trained to align the point cloud representation $\boldsymbol{z}_i^P = f^P(\boldsymbol{x}_i^P)/||f^P(\boldsymbol{x}_i^P)||$ to the CLIP embedding space. In this case, the optimization objective becomes:

$$\frac{1}{2}(\mathcal{L}^{P\leftrightarrow I}(\boldsymbol{x}_i^P, \boldsymbol{x}_i^I; f^P, f^I, \tau) + \mathcal{L}^{P\leftrightarrow T}(\boldsymbol{x}_i^P, \boldsymbol{x}_i^T; f^P, f^T, \tau)) \tag{2}$$

## 3.2 REVISITING TRAINING RECIPE

It is known to the 3D community that a well-tuned training recipe brings a dramatic performance boost (Qian et al., 2022). Yet, despite its astonishingly good performance, the training recipe of the image-text-3D contrastive learning paradigm is underexplored. Thus, before diving deep into our proposed method, we first revisit the training recipe of ULIP (Xue et al., 2023a) and OpenShape (Liu et al., 2023a), identifying several changes that significantly bolster 3D representation learning.

- **Batchsize**. Contrastive learning benefits significantly from a large batch size (Cherti et al., 2023; Radford et al., 2021)). Nevertheless, state-of-the-art model (Xue et al., 2023a) still adopts a batchsize as low as 64. In this regard, we find that a medium batchsize of 2k strikes a good trade-off between different datasets. Interestingly, an even larger batchsize does not lead to notable improvement, presumably because of the limited data scale and frozen CLIP encoders. Thus, we opt for a batchsize of 2k by default in this paper, unless otherwise noted.

- **Learning rate schedule**. Different from ULIP, OpenShape adopts the step learning rate decay schedule without warmup. Yet, the default setting in CLIP (Radford et al., 2021; Cherti et al., 2023) is to adopt the cosine learning rate decay schedule with the warmup, a setting known to help better train an image model (He et al., 2019). We adopt the same setting as CLIP and find it leads to clear improvement.

- **Exponential moving average**. During training, we observe that the model performance steadily increased on the synthetic Objaverse-LVIS dataset, while fluctuating drastically on the real-scanned ScanObjectNN dataset, presumably due to the domain gap. To alleviate the fluctuation issue, we employ Exponential Moving Average (EMA) (Tarvainen & Valpola, 2017) to stabilize training.

- **Separate temparature**. Features from different modalities may have different distributions. Prior works (Xue et al., 2023a; Liu et al., 2023a) use a shared temperature parameter $\tau$ (Wu et al., 2018) to control the level of concentration of multi-modal features. By contrast, using separate temperature parameters for each modality results in better performance.

The training recipe differences are summarized in Table 1. Together, our enhanced recipe drastically boosts the top-1 accuracy of the OpenShape baseline by 3.3%, 3.6%, and 1.9% on the Objaverse-LVIS, ScanObjectNN, and ModelNet40 datasets, respectively, demonstrating the huge potential of a well-tuned training recipe. Next, we introduce two enhancements designed specifically for 3D contrastive learning in detail.

## 3.3 IMAGE-3D TO TEXT JOINT REPRESENTATION ALIGNMENT

3D point cloud and 2D images are known to encode different yet complimentary cues: point cloud scans better capture depth and geometry information, while images better capture dense semantic information (Bai et al., 2022; Liu et al., 2023b; Chen et al., 2023; Wang et al., 2023). Intuitively, the

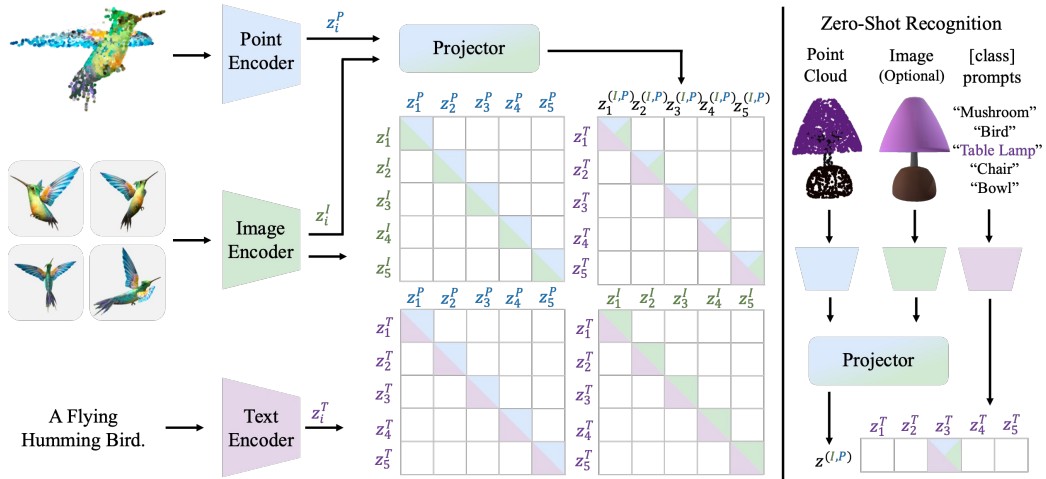

Figure 2: Summary of our MixCon3D framework. MixCon3D first extracts the representation of input triplets (images, text, point cloud) from a pre-trained vision-language model (*e.g.*, CLIP) and a 3D encoder (*e.g.*, PointBERT). Then the image and point cloud features go through a projector to obtain the joint modal features, serving as complementary representation. The contrastive losses are applied to align features among three modalities (image-text-3D) and joint representation to text.

fusion of information from these two modalities could more accurately represent the corresponding object in the real world, and thus achieve better alignment with the text modality. In this section, we propose a simple yet effective image-3D to text joint representation alignment approach, which constructs a new joint image and point cloud representation by aggregating the features extracted from these modalities. In addition to contrast the conventional tri-modal features with each other, the joint representation will also be aligned with text features via an additional constrastive loss.

Specifically, given batched data triplets $\{x_i = (x_i^I, x_i^T, x_i^P)\}_{i=1}^N$ and image-text-3D models ($f^I$, $f^T$, $f^P$), the corresponding features are denoted as $\mathbb{R}^D$ vectors ($z_i^I$, $z_i^T$, $z_i^P$), respectively. To model the joint representation, we concatenate the image features and point cloud features (*i.e.*, concat($z_i^I, z_i^P$) $\in \mathbb{R}^{2\times D}$), and use a linear layer $g^{(I,P)}$ to project the joint representation to a suitable dimension $z_i^{(I,P)} = g^{(I,P)}(\text{concat}(z_i^I, z_i^P))$. The extra contrastive term is as follows:

$$\mathcal{L}^{(I,P)\leftrightarrow T}(x_i^I, x_i^P, x_i^T) = -\frac{1}{2N}\sum_i^N(\log\frac{\exp(z_i^{(I,P)}\cdot z_i^T/\tau)}{\sum_j\exp(z_i^{(I,P)})\cdot z_j^T/\tau)} + \log\frac{\exp(z_i^T\cdot z_i^{(I,P)}/\tau)}{\sum_j\exp(z_i^T\cdot z_i^{(I,P)}/\tau)})$$

(3)

where $\tau$ is the temperature parameter. In this case, the overall objective becomes:

$$\mathcal{L}^{(I,P)\leftrightarrow T}(x_i^I, x_i^P, x_i^T) + \frac{1}{3}(\mathcal{L}^{P\leftrightarrow I}(x_i^P, x_i^I) + \mathcal{L}^{P\leftrightarrow T}(x_i^P, x_i^T) + \mathcal{L}^{I\leftrightarrow T}(x_i^I, x_i^T))$$

(4)

where $\{x_i = (x_i^I, x_i^T, x_i^P)\}_{i=1}^N$ is the input image-text-point cloud data triplets.

Note that in Equation 4, the conventional point cloud to text loss $\mathcal{L}^{I\leftrightarrow T}$ is kept. This enables the model to make predictions solely based on 3D input even when corresponding images are unavailable, as in ScanObjectNN (Uy et al., 2019) or ModelNet40 (Wu et al., 2015). Even so, we observe a notable performance boost brought by the extra term $\mathcal{L}^{(I,P)\leftrightarrow T}$, presumably better aligned point cloud and text representations, shown as in Section 4.3

Also, different from the ULIP or OpenShape practice (Xue et al., 2023a; Liu et al., 2023a), we add another image to text alignment $\mathcal{L}^{P\leftrightarrow T}$ and an additional learnable projection head upon the frozen CLIP encoder. Intriguingly, as shown in Section 4.3, without our joint alignment loss, we find this $\mathcal{L}^{P\leftrightarrow T}$ loss even leads to performance drop. On the contrary, with our joint alignment loss, we find $\mathcal{L}^{P\leftrightarrow T}$ leads to even further improvement, highlighting the critical effect of image and point cloud representation fusion.

## 3.4 Synergy with the Multi-view Mechanism

Even with joint representation alignment, the fused image and point cloud features are still not optimal for 3D object description, because a single-view image only contains information captured from a specific angle. Multi-view based methods, a prominent category of 3D representation approaches, have demonstrated promising performance in 3D understanding tasks (Su et al., 2015; Jaritz et al., 2019; Hamdi et al., 2021; 2023). Interestingly, though previous works like ULIP (Xue et al., 2023a;b) and OpenShape (Liu et al., 2023a) render images from multiple viewpoints of the same point cloud when creating the data triplets, they merely sample one image from the rendered multi-view images when extracting the image features, which inherently encode only partial facets of the 3D object. In this paper, we propose to capitalize on the features accumulated from multi-view images to achieve a more holistic view of a 3D object. Specifically, given a set of multi-view images $\boldsymbol{x}_i^I = \{\boldsymbol{x}_{(i,j)}^I\}_{j=1}^M$, which corresponds to the text description $\boldsymbol{x}_i^T$, and point cloud $\boldsymbol{x}_i^P$, we propose to replace the single-view image feature $\boldsymbol{z}_i^I$ with the fusion of features extracted from individual features $\boldsymbol{z}_{(i,j)}^I$. Note that $\boldsymbol{z}_i^{(I,P)}$ is changed accordingly. Specifically, we simply adopt view-pooling (Su et al., 2015) to aggregate the multi-view information from $\boldsymbol{x}_{(i,j)}^I$.

## 4 Experiments

We first introduce the experimental setup in Section 4.1, including different pre-training datasets, downstream benchmarks, and implementation details. Our main results are presented in Section 4.2, where we show that our MixCon3D beats prior methods by a large margin with various backbones and pre-training datasets. In Section 4.3, we conduct a series of ablation studies on the key components in the MixCon3D. Additionally, we establish the applicability of 3D representation learned by MixCon3D in the realm of cross-modal applications such as text to 3D object retrieval and point cloud captioning (Section 4.4).

## 4.1 Experimental Setup

**Pre-training datasets.** Following OpenShape (Liu et al., 2023a), the full pre-training dataset (denoted as "Ensemble") encompasses four prominent datasets: ShapeNet (Chang et al., 2015), 3D-FUTURE (Fu et al., 2021), ABO (Collins et al., 2022)) and Objaverse (Deitke et al., 2023). The position of the point cloud is obtained by sampling 10,000 points from the mesh surface and the color is interpolated based on the mesh textures. The images are rendered from 12 preset camera poses that cover the whole object uniformly. Then, the paired texts are generated by BLIP (Li et al., 2022; 2023) and Azure cognition services with GPT4 (OpenAI, 2023) to filter out noisy text. Readers are encouraged to refer to Liu et al. (2023a) for more details. Besides, we also verify the effectiveness of our method trained by the ShapeNet dataset only and the ensembled dataset except for the LVIS (Gupta et al., 2019) categories (denoted as "Ensemble (No LVIS)").

**Down-stream datasets.** The following three datasets are used for the evaluation of zero-shot point cloud recognition. (a) ModelNet40 (Wu et al., 2015) is a synthetic dataset comprising 3D CAD models, including 9,843 training samples and 2,468 testing samples, distributed across 40 categories. (b) ScanObjectNN (Uy et al., 2019) is a dataset composed of 3D objects acquired through real-world scanning techniques, encompassing a total of 2,902 objects that are systematically categorized into 15 distinct categories We follow (Xue et al., 2023a;b; Liu et al., 2023a) and use the variants provided by (Yu et al., 2022) in our experiments. (c) Objaverse-LVIS, an annotated subset of the Objaverse (Deitke et al., 2023), incorporates a corpus of 46,832 shapes originating from 1,156 categories as delineated in the LVIS dataset (Gupta et al., 2019).

**Implementation details.** We implement our approach in PyTorch (Paszke et al., 2019) and train the models on a server with 8 NVIDIA A5000 GPUs with a batch size of 2048. We train the model for 200 epochs with the AdamW (Loshchilov & Hutter, 2018) optimizer, a warmup epoch of 10, and a cosine learning rate decay schedule (Loshchilov & Hutter, 2016). The base learning rate is set to 1e-3, based on the linear learning rate scaling rule (Goyal et al., 2017): $lr = base\_lr \times$ batchsize / 256. The EMA factor is set to 0.9995. Following Liu et al. (2023a), OpenCLIP ViT-bigG-14 (Cherti et al., 2023) is adopted as the pretrained CLIP model. As for the point tokenization, we follow Yu et al. (2022) to partition the points into 512 point groups (subclouds), with subcloud containing

Table 2: Comparison with state-of-the-art methods on three representative zero-shot 3D reognition benchmarks. "Top1-C" means the top-1 class average accuracy. "Encoder" denotes the point cloud encoder used in the framework.

| Method | Encoder | Training data | Objaverse-LVIS | | | | ScanObjectNN | | | | ModelNet40 | | | |
|---|---|---|---|---|---|---|---|---|---|---|---|---|---|---|
| | | | Top1 | Top1-C | Top3 | Top5 | Top1 | Top1-C | Top3 | Top5 | Top1 | Top1-C | Top3 | Top5 |
| PointCLIP | - | Depth inference | 1.9 | - | 4.1 | 5.8 | 10.5 | - | 20.8 | 30.6 | 19.3 | - | 28.6 | 34.8 |
| PointCLIP v2 | - | | 4.7 | - | 9.5 | 12.9 | 42.2 | - | 63.3 | 74.5 | 63.6 | - | 77.9 | 85.0 |
| ReCon | - | ShapeNet | 1.1 | - | 2.7 | 3.7 | 61.2 | - | 73.9 | 78.1 | 42.3 | - | 62.5 | 75.6 |
| CG3D | - | | 5.0 | - | 9.5 | 11.6 | 42.5 | - | 57.3 | 60.8 | 48.7 | - | 60.7 | 66.5 |
| CLIP2Point | - | | 2.7 | - | 5.8 | 81.2 | 25.5 | - | 44.6 | 59.4 | 49.5 | - | 71.3 | 81.2 |
| ULIP | PointBERT | | 6.2 | - | 13.6 | 17.9 | 51.5 | - | 71.1 | 80.2 | 60.4 | - | 79.0 | 84.4 |
| OpenShape | SparseConv | | 11.6 | - | 21.8 | 27.1 | 52.7 | - | 72.7 | **83.6** | 72.9 | - | 87.2 | 93.0 |
| MixCon3D | SparseConv | | **23.5** | **17.5** | **40.2** | **47.1** | **54.4** | **56.1** | 73.9 | 83.3 | **73.9** | **70.2** | **88.2** | **94.0** |
| OpenShape | PointBERT | | 10.8 | - | 20.2 | 25.0 | 51.3 | - | 69.4 | 78.4 | 70.3 | - | 86.9 | 91.3 |
| MixCon3D | PointBERT | | 22.3 | 16.2 | 37.5 | 44.3 | 52.6 | 52.1 | 69.9 | 78.7 | 72.6 | 68.2 | 87.1 | 91.3 |
| ULIP | PointBERT | Ensemble (No LVIS) | 21.4 | - | 38.1 | 46.0 | 46.0 | - | 66.1 | 76.4 | 71.4 | - | 84.4 | 89.2 |
| OpenShape | SparseConv | | 37.0 | - | 58.4 | 66.9 | 54.9 | - | 76.8 | 87.0 | 82.6 | - | 95.0 | 97.5 |
| MixCon3D | SparseConv | | 45.7 | 33.5 | 67.0 | 73.2 | 56.5 | 60.5 | 77.8 | 87.5 | 83.3 | 82.4 | 95.6 | 97.6 |
| OpenShape | PointBERT | | 39.1 | - | 60.8 | 68.9 | 47.2 | - | 72.4 | 84.7 | 85.3 | - | 96.2 | 97.4 |
| MixCon3D | PointBERT | | **47.5** | **34.6** | **69.0** | **76.2** | **57.7** | **61.5** | **80.7** | **89.8** | **87.3** | **86.7** | **96.8** | **98.1** |
| ULIP | PointBERT | Ensemble | 26.8 | - | 44.8 | 52.6 | 51.6 | - | 72.5 | 82.3 | 75.1 | - | 88.1 | 93.2 |
| OpenShape | SparseConv | | 43.4 | - | 64.8 | 72.4 | 56.7 | - | 78.9 | 88.6 | 83.4 | - | 95.6 | 97.8 |
| MixCon3D | SparseConv | | 47.3 | 35.0 | 68.7 | 76.1 | 57.1 | 61.2 | 79.2 | 88.9 | 83.9 | 83.2 | 95.9 | 98.0 |
| OpenShape | PointBERT | | 46.8 | 34.0 | 69.1 | 77.0 | 52.2 | 53.2 | 79.7 | 88.7 | 84.4 | 84.9 | 96.5 | 98.0 |
| MixCon3D | PointBERT | | **52.5** | **38.8** | **74.5** | **81.2** | **58.6** | **62.3** | **80.3** | 89.2 | **86.8** | **86.8** | **96.9** | **98.3** |

precisely 32 points. Then, a mini-PointNet (Qi et al., 2017a) is adopted to project those sub-clouds into point embeddings.

## 4.2 MAIN RESULTS

In Table 2, we compare the performance of our MixCon3D with state-of-the-art competitors across two representative encoders, SparseConv (Choy et al., 2019) and PointBERT (Yu et al., 2022); three different training set, "ShapeNet", "Ensemble (No LVIS)", and "Ensemble"; and three popular 3D recognition benchmarks, Objaverse-LVID, ScanObjectNN, and ModelNet40. We observe that our MixCon3D consistently exhibits superior performance. Specifically, on the challenging long-tailed benchmark Objaverse-LVIS, MixCon3D greatly improves the zero-shot Top1 accuracy from 46.8% of OpenShape to **52.5%** with PointBERT encoder and "Ensemble" training data. Besides, when tested on the ScanObjectNN dataset that comprises scanned points of real objects and thus a bigger domain gap (Uy et al., 2019), our MixCon3D also achieves a significant performance boost of **6.4%** (58.6% *v.s.* 52.2%), indicating the generalizability of our learned 3D representation. These results validate the effectiveness of our proposed MixCon3D, demonstrating a more powerful open-world 3D understanding ability.

## 4.3 ABLATION STUDIES

**Improved training recipe** We show the effect of improved training strategies in Table 3. For example, the separate temperature setup obtains a notable performance improvement of 0.8% on ScanObjectNN. A larger batchsize is beneficial to image-text-3D contrastive pre-training on all three datasets, where Objaverse-LVIS shows the largest improvement of 1.2%. We observe a similar effect of the cosine learning rate schedule with warmup. Lastly, the exponential moving average update brings at least a 1% performance increase on all three datasets.

**MixCon3D component.** In Table 4, we analyze the effect of each critical component in MixCon3D. Interestingly, we find that the image-text alignment alone even leads to worse performance compared to the baseline, which is probably why ULIP (Xue et al., 2023a) and OpenShape (Liu et al., 2023a)

Table 3: Ablation studies for sequentially applying the improved training strategies for constructing a strong baseline on downstream zero-shot tasks.

| Improvements | Objaverse-LVIS | | | | ScanObjectNN | | | | ModelNet40 | | | |
|---|---|---|---|---|---|---|---|---|---|---|---|---|
| | Top1 | Top1-C | Top3 | Top5 | Top1 | Top1-C | Top3 | Top5 | Top1 | Top1-C | Top3 | Top5 |
| OpenShape | 46.5 | 34.0 | 69.0 | 76.8 | 52.0 | 53.2 | 77.5 | 87.5 | 84.2 | 84.9 | 95.9 | 97.4 |
| + Separate Temperature | 46.8 | 34.4 | 69.2 | 77.1 | 52.8 | 54.0 | 77.6 | 87.4 | 84.4 | 84.6 | 96.1 | 97.4 |
| + Large Batchsize | 48.0 | 35.3 | 70.1 | 77.4 | 53.5 | 55.5 | 78.0 | 87.7 | 84.8 | 85.3 | 96.4 | 97.7 |
| + LR Schedule | 48.5 | 36.0 | 70.6 | 77.7 | 54.1 | 56.3 | 78.2 | 87.9 | 85.0 | 85.0 | 96.4 | 97.9 |
| + EMA | 49.8 | 36.9 | 71.7 | 78.7 | 55.6 | 58.9 | 79.3 | 88.6 | 86.1 | 86.2 | 96.8 | 98.3 |

Table 4: The ablation studies of different components in the proposed MixCon3D.

| $\mathcal{L}^{I \leftrightarrow T}$ | $\mathcal{L}^{(I,P) \leftrightarrow T}$ | Multi-View | Objaverse-LVIS | | | | ScanObjectNN | | | | ModelNet40 | | | |
|---|---|---|---|---|---|---|---|---|---|---|---|---|---|---|
| | | | Top1 | Top1-C | Top3 | Top5 | Top1 | Top1-C | Top3 | Top5 | Top1 | Top1-C | Top3 | Top5 |
| ✗ | ✗ | ✗ | 49.8 | 36.9 | 71.7 | 78.7 | 55.6 | 58.9 | 79.3 | 88.6 | 86.1 | 86.2 | 96.8 | **98.3** |
| ✓ | ✗ | ✗ | 48.7 | 36.2 | 70.4 | 77.7 | 55.4 | 59.7 | 75.8 | 85.6 | 84.7 | 84.8 | 96.6 | 97.9 |
| ✗ | ✓ | ✗ | 51.0 | 37.8 | 73.2 | 79.5 | 57.9 | 61.4 | 79.8 | **89.3** | 86.5 | 86.4 | 96.6 | 98.0 |
| ✓ | ✓ | ✗ | 51.6 | 38.2 | 73.7 | 80.6 | 58.1 | 61.9 | **80.3** | 89.2 | 86.6 | 86.6 | 96.4 | 98.1 |
| ✓ | ✓ | ✓ | **52.5** | **38.8** | **74.5** | **81.2** | **58.6** | **62.3** | **80.3** | 89.2 | **86.8** | **86.8** | **96.9** | **98.3** |

didn't employ the image-text contrastive loss $\mathcal{L}^{I \leftrightarrow T}$. By contrast, our proposed image-3D to text joint alignment loss $\mathcal{L}^{(I,P) \leftrightarrow T}$ itself brings a considerable performance boost of at least 1.8% on all three datasets, and combining it $\mathcal{L}^{I \leftrightarrow T}$ leads to further improvement. This clearly shows the paramount importance of aggregating complimentary useful cues in the contrastive learning with image, point cloud, and text. Another crucial component is extracting multi-view image features, which results in a further improvement of 0.9% on the Objaverse-LVIS dataset.

**Multi-modal Inference.** The introduction of joint alignment and multi-view images leads to a lot of inference options. For instance, whether we should use use point cloud input alone, or combine point cloud with image input. Also, whether to apply a single-view image or multi-view images. We ablate a series of inference ways that aggregate different representations, and show the results in Table 5. As can be seen, the way of point cloud and image representation fusion, plus multi-view image feature extraction, surpasses all other options by a clear margin, underpinning the significance of knowledge aggregation from different representations.

Table 5: The analysis of cross-modal representation ensembing schemes in Objaverse-LVIS.

| Point Cloud | Image | Multi-View | Objaverse-LVIS | | | |
|---|---|---|---|---|---|---|
| | | | Top1 | Top1-C | Top3 | Top5 |
| ✓ | ✗ | - | 50.4 | 37.4 | 72.2 | 79.1 |
| ✗ | ✓ | - | 44.5 | 34.5 | 64.2 | 70.6 |
| ✗ | ✓ | ✓ | 51.9 | 38.5 | 73.1 | 79.4 |
| ✓ | ✓ | - | 51.6 | 37.6 | 73.4 | 80.1 |
| ✓ | ✓ | ✓ | **52.5** | **38.8** | **74.5** | **81.2** |

## 4.4 CROSS-MODAL APPLICATIONS

To test how well the point cloud representation of our MixCon3D is aligned with CLIP pre-trained representations, we evaluate the learned representations on the following cross-modal tasks, following the practice in Liu et al. (2023a).

**Text to 3D object retrieval.** We use cosine similarity between text embeddings of a specific input and 3D shape embeddings from the ensembled dataset as the ranking metric. We compare the retrieval result of our MixCon3D with that of OpenShape. As shown in Figure 3, our MixCon3D can capture more comprehensive feature representation, *e.g.*, allowing for more accurate indexing such as the hummingbird and fine-grained retrieval in situations where the "lamp" is required to have an "electric wire".

Input Text    "A blue table lamp with electric wire"      "A beautiful hummingbird"

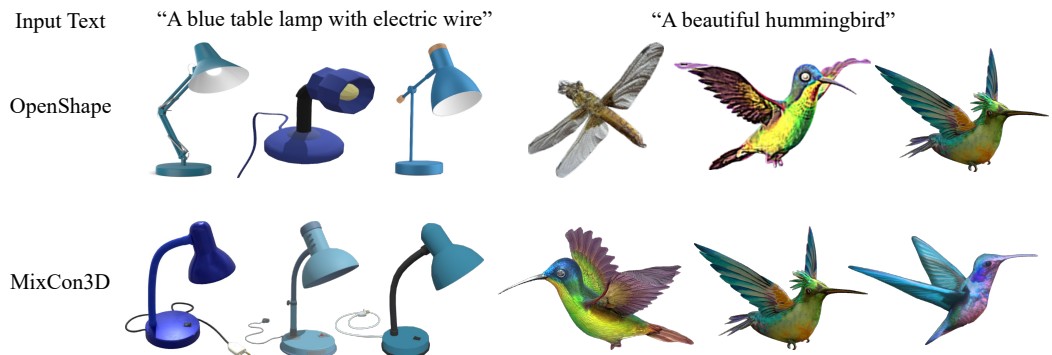

Figure 3: **Text to 3D object retrieval comparisons.** The input text and the first three retrieved 3D objects are listed in each column for both OpenShape and our MixCon3D. Compared to OpenShape, the 3D representation learned by our MixCon3D enables more accurate and finer-grained retrieval tasks.

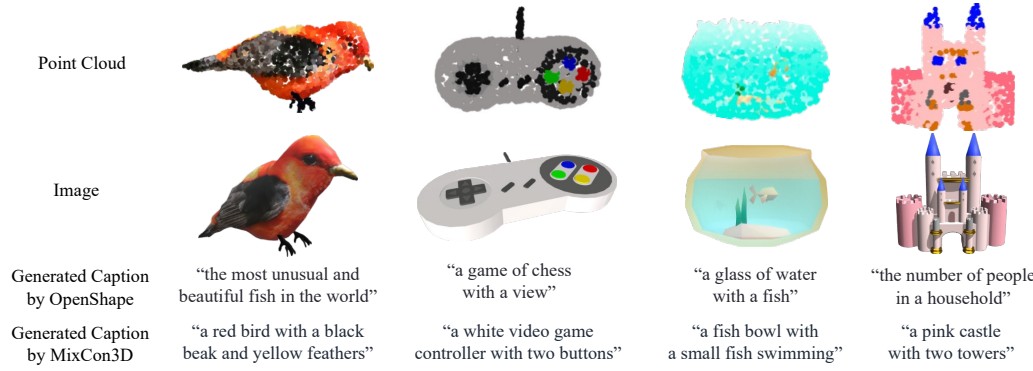

Figure 4: **Point cloud captioning comparisons.** In each row, we list the input point cloud, corresponding images, and generated captions by OpenShape and MixCon3D. The generated captions indicate that our method is capable of extracting a more comprehensive 3D representation.

**Point cloud captioning.** We feed the 3D shape embeddings of our MixCon3D into an off-the-shelf image captioning model ClipCap (Mokady et al., 2021), and compare the results with that of OpenShape. As can be observed in Figure 4, our MixCon3D captures a better 3D shape representation, thereby enhancing the generative model's ability to generate more accurate and comprehensive captions.

## 5 CONCLUSION

In this paper, we present MixCon3D, a simple yet effective image-text-3D contrastive learning approach, which synergizes multi-modal joint alignment and multi-view representations for better open-world 3D understanding capability. Specifically, we propose to construct a simple yet effective image-3D to text joint representation alignment training scheme and capitalize on the features accumulated from multi-view images. We also provide a first detailed training guideline in the field of image-text-3D contrastive learning. Together with the improved training pipeline, MixCon3D not only achieves superior performance on a wide range of 3D recognition benchmarks but also facilitates downstream cross-modal applications such as text to 3D object retrieval and point cloud captioning. We hope that our work could encourage more research endeavors in building the next-generation open-world 3D model.

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
