# OpenReview forum: "MIXCON3D: SYNERGIZING MULTI-VIEW AND CROSS-MODAL CONTRASTIVE LEARNING FOR ENHANCING 3D REPRESENTATION"
_ICLR.cc/2024/Conference — ICLR 2024 Conference Withdrawn Submission_

### Official Review · Reviewer_EuDk · 2023-10-30

**Soundness:** 2 fair
**Presentation:** 2 fair
**Contribution:** 2 fair
**Rating:** 5
**Confidence:** 5

**Summary:**

This paper presents MixCon3D, trying to explore the synergy between multi-view representation and cross-modal contrastive learning in open-world 3D representation learning. The main idea is to leverage a joint representation alignment that aligns fused multimodal representations among image, text, and point cloud triplet signals. Besides, multi-view input is used to facilitate 2D-based 3D understanding further. Experiments on classical 3D zero-shot learning, 3D content retrieval, and 3D captioning show promising results.

**Strengths:**

- The proposed method is very simple but effective and can be easily combined with existing methods. Experimental results demonstrate a strong improvement.
- The idea of exploring the synergy between multi-view 2D representation to facilitate 3D understanding during cross-modal contrastive learning is well-motivated and somewhat novel to me.

**Weaknesses:**

- **Major:** The proposed method is somewhat like some tricks that are incremental to me. The proposed joint alignment is very simple and makes sense, but some in-depth analysis is missing. For example, what is the synergy that is demonstrated? The paper claims that `dense semantic information` is within the 2D representations, while multi-view can further `fortifying cross-modal alignment`. The ablation studies is good, but it would be better if the authors could show some interesting and intuitive analysis that demonstrates what synergy is enhanced.
- **Comparison:** To the best of my knowledge, the proposed method is similar to GMC [1] and MCLEA [2], which also propose cross-modal contrastive learning between fused multimodal representations and single-modal representations, can authors explain the differences?
- **Suggestions:** i) It would be useful if the authors could provide the `validation loss curves` comparisons of different learning settings with baseline OpenShape to demonstrate the generalization improvements with the proposed techniques like joint representation alignment; ii) A `visualization of learned association` between 2D representation and 3D representation would be interesting.
- The improved training recipe is effective, but the insight behind this strategy is limited. It looks like a combination of useful tricks while a large amount of space is used to introduce these strategies. In this case, with all experiments showing benchmarking results, the current paper looks somewhat like an experimental report rather than a scientific paper.

**Questions:**

- What is the underlying reason why the joint representation alignment is better? Can authors provide more in-depth explanations? It would be better if the authors could show some discussion experiments.
- Why only show the results of OpenShape with the proposed training recipe? What about the ablation results of MixCon3D with the training recipe?
- Besides zero-shot results, I wonder about the supervised fine-tuning results on downstream tasks like classification on ScanObjectNN. Make a comparison to methods including single-modal methods Point-BERT [3], Point-MAE [4], and cross-modal methods ACT [5], ReCon [6], and I2P-MAE [7].
- Missing related work discussions. Cross-modal 3D representation learning methods ACT [5], I2P-MAE [7], CrossPoint [8], SimIPU [9], P2P [10] and Pix4Point [11]. Cross-modal open-world 3D scene understanding methods OpenScene [12] and CLIP-FO3D [13].
- Typo. i) In Figure 1, the proposed MixCon3D should be red. ii) All Quotation marks should be “; currently, all of these marks are close quotations. (In Latex, it should be ``) iii) On page 5, after Eq (4), "the conventional point cloud to text loss $\mathcal{L}^{I\leftrightarrow T}$" should be "... $\mathcal{L}^{P\leftrightarrow T}$", right? iv) There are some other small typos, just name a few here.
- It is better if the paper title on the OpenReview system is not written all in capital letters, just write the title as it is with the first letters capitalized.

Although somewhat incremental, overall, I think the proposed method has good motivation and is very simple and effective. I would be happy to increase my rating if the authors could address my concerns to make the paper presentation more solid and interesting, especially my major concerns and the clarification on comparisons to previous literature.

[1] Geometric Multimodal Contrastive Representation Learning. In ICML 2022.\
[2] Multi-modal Contrastive Representation Learning for Entity Alignment. In COLING 2022.\
[3] Point-BERT: Pre-training 3D Point Cloud Transformers with Masked Point Modeling. In CVPR 2022.\
[4] Masked Autoencoders for Point Cloud Self-supervised Learning. In ECCV 2022.\
[5] Autoencoders as Cross-Modal Teachers: Can Pretrained 2D Image Transformers Help 3D Representation Learning? In ICLR 2023.\
[6] Contrast with Reconstruct: Contrastive 3D Representation Learning Guided by Generative Pretraining. In ICML 2023.\
[7] I2P-MAE: Learning 3D Representations from 2D Pre-trained Models via Image-to-Point Masked Autoencoders. In CVPR 2023.\
[8] CrossPoint: Self-Supervised Cross-Modal Contrastive Learning for 3D Point Cloud Understanding. In CVPR 2022.\
[9] SimIPU: Simple 2D Image and 3D Point Cloud Unsupervised Pre-Training for Spatial-Aware Visual Representations. In AAAI 2022.\
[10] P2P: Tuning Pre-trained Image Models for Point Cloud Analysis with Point-to-Pixel Prompting. In NeurIPS 2022.\
[11] Improving Standard Transformer Models for 3D Point Cloud Understanding with Image Pretraining. In 3DV 2023.\
[12] OpenScene: 3D Scene Understanding with Open Vocabularies. In CVPR 2023.\
[13] CLIP-FO3D: Learning Free Open-world 3D Scene Representations from 2D Dense CLIP. In ICCVW 2023.

---

### Official Review · Reviewer_QgYi · 2023-11-04

**Soundness:** 3 good
**Presentation:** 3 good
**Contribution:** 2 fair
**Rating:** 5
**Confidence:** 5

**Summary:**

In this paper, the authors propose MixCon3D for 3D cross-modal contrastive learning. The authors introduce the multi-view images to interact with the 3D features for dense semantic information. In addition, authors ensemble multiply datasets for contrastive learning, which assists models to get the state-of-the-art on the zero-shot in the downstream dataset. The additional quantitative results of image retrieval and point cloud caption show the generalization.

**Strengths:**

1. The paper is well-written and easy to understand. The approaches are easy to follow.
2. Experiments state that the method achieves state-of-the-art performance, and the quantitative shows great robustness on the downstream tasks.
3. The motivation is clear and reasonable.

**Weaknesses:**

1. Additional Comparisons:
Your zero-shot classification experiments on the ModelNet40 and other datasets (Table 2) offer valuable insights. To further substantiate your findings, it could be beneficial to include comparisons with some additional methods that have shown promise in this domain, such as JM3D [1] and ULIP-2 [2]. Such comparisons could provide a more comprehensive view of how your approach stands in the context of the existing state-of-the-art methods and highlight the unique contributions of your work more effectively.

2. Clarification of Contribution:
You have noted the introduction of multi-view images in the context of contrastive learning across multiple modalities as a key contribution to your paper. It is indeed an interesting aspect of your research. However, to strengthen your claim, it would be constructive to discuss how your approach differentiates from or builds upon prior works like JM3D [1] and ULIP-2 [2], which have also explored multi-view images within a similar framework. Clarifying this could help readers appreciate the novel aspects of your methodology and its significance in the field.

3. Expansion of Quantitative Experiments:
The paper presents intriguing insights into the arithmetic of multi-modalities. To solidify these findings, it might be valuable to expand on the quantitative experiments. Additional data or more varied testing conditions could offer deeper validation of your hypothesis and might also uncover further interesting aspects of your model's behavior. This would not only bolster your current claims but could also provide a richer resource for readers interested in the intricacies of multi-modal contrastive learning.

[1] Beyond First Impressions: Integrating Joint Multi-modal Cues for Comprehensive 3D Representation, ACM MM 23

[2] ULIP-2: Towards Scalable Multimodal Pre-training For 3D Understanding, arXiv: 2305.08275

**Questions:**

1. Multi-View Analysis: Could you specify the number of views used during training and consider including an ablation study on this feature? This would greatly clarify the contribution of the multi-view approach.

2. Extended Experiments: Exploring fine-grained tasks like semantic segmentation, as in ULIP and JM3D, could offer deeper insights and substantiate your model’s versatility.

3. Clarification on Table 5: Please clarify the meaning of "no point cloud" in rows 2 and 3, to better understand how these metrics are compared without point cloud inputs.

4. Typographical Corrections: I've noticed a minor typographical error on the 10th row of the 8th page, where the word "use" is repeated.

---

### Official Review · Reviewer_7QFb · 2023-11-08

**Soundness:** 1 poor
**Presentation:** 2 fair
**Contribution:** 1 poor
**Rating:** 3
**Confidence:** 4

**Summary:**

The paper presents MixCon3D, an approach that extends the success of Contrastive Language-Image Pre-training to the 3D domain. It introduces a joint representation alignment mechanism, enriching tri-modal representations by combining images and point clouds. Extensive experiments on multiple benchmarks demonstrate a 5.7% performance gain over the previous state-of-the-art on the Objaverse-LVIS dataset. The approach is also effective in text-to-3D retrieval and point cloud captioning applications.

**Strengths:**

1. The paper conducts experiments across various tasks, including zero-shot classification, retrieval, and captioning.
2. The paper is well-written, with a clear and concise methodology.

**Weaknesses:**

1. Inequitable comparisons, MixCon3D employs ground truth rendered images as input data in zero-shot inference, which is certain to enhance its semantic comprehension performance.
2. Compared to ULIP&ULIP-2 [Liu et al., 2023], the improvements in the paper are incremental, with the sole modification being the introduction of $L^{(I, P)↔T}$.
3. The results for some methods in Table 2, such as ReCon, are inaccurate. The results of ScanObjectNN and ModelNet are reversed.

[Liu et al., 2023] ULIP-2: Towards Scalable Multimodal Pre-training for 3D Understanding. arXiv preprint.

**Questions:**

See Weaknesses.

---

### Official Review · Reviewer_Kf2T · 2023-11-09

**Soundness:** 2 fair
**Presentation:** 2 fair
**Contribution:** 2 fair
**Rating:** 5
**Confidence:** 4

**Summary:**

This paper is focused on adapting pretrained vision-language models for point-centric perception tasks, especially in shape classification. Compared with previous works, this paper delivers impressive improvements even on the large-scale Objaverse-LVIS dataset. The authors also provide some quantitative analysis of their methods on text-point retrieval and point cloud caption tasks.

**Strengths:**

* This paper provides a very strong performance on the Objaverse-LVIS dataset, which is promising and convincing.
* This paper provides detailed training strategies, which is helpful for further studies.
* Besides training strategies, this paper also offers very detailed ablation study results.
* The proposed is very easy to understand and follow.

**Weaknesses:**

* **Technical Novelty**. Although this paper delivers impressive improvement in the area, the technical contribution is limited. The insights of this paper seem more focused on how to train a good model instead of solving a problem. For instance, in the introduction part, the authors wrote "synergy is underexplored, with each modality isolated in contrastive learning", after reading the whole paper, I still do not understand why this paper is able to perform multimodal synergy compared with ULIP, OpenShape, or other relevant methods. The difference is multi-view contrastive learning? How images and point clouds are synergized in the proposed method? Considering the method itself, multi-view projection between point cloud and images is directly adopted from existing methods, based on more visual clues, the proposed method offers improvement to baselines. Consequently, there are more constrastive pairs for learning objectives. However, I prefer this paper as a well-written engineering blog instead of an academic publication. And I sincerely appreciate the contributioins in this paper.

* **Experimental Designs**. An important ablation study. As shown in Table 4, with proposed training strategies, the method achieves 49.8% Acc. on the Objaverse-LVIS dataset. Introducing $\mathcal{L}^{T-I}$ holds back the performance by -0.9%. Why the loss function are not combined with multi-view projection? More specifically,  why the performance of $\mathcal{L}^{T-I}$ and multi-view projection is not reported? In addition to this, the ablation seeting on the number of views projected to image is not ablated either. It's highly suggested to report the number of views on pretraining process. Is the optimal setting to project 4 views by 90 degress?

* Moreover, there are also shortcomings in this paper. The visualization results seem not supportive of the corresponding arguments. In Figure 3, the authors provide visualization results on text-point retrieval experiments. These results are top-3 ranking retrival results. For the left part, I do not think the proposed method significantly outperform OpenShape. Of cource, if the figure gets revised on top-10 or top-5 retrieval results, it would be more convincing.

**Questions:**

See weakness.

**Details Of Ethics Concerns:**

None.